# Antifungals and Drug Resistance

Chowdhury Mobaswar Hossain [1,*] , Lisa Kathleen Ryan [2,3] , Meeta Gera [4], Sabyasachi Choudhuri [1] , Nazmun Lyle [5], Kazi Asraf Ali [1] and Gill Diamond [2]

1   Department of Pharmaceutical Technology, Maulana Abul Kalam Azad University of Technology, Haringhata 741249, India
2   Department Oral Immunology and Infectious Diseases, School of Dentistry, University of Louisville, Louisville, KY 40202, USA
3   Department of Medicine, Division of Infectious Diseases and Global Medicine, University of Florida, Gainesville, FL 32669, USA
4   Department of Biochemical Engineering and Biotechnology, Indian Institute of Technology, New Delhi 110016, India
5   International Institute of Innovation and Technology, Kolkata 700156, India
*   Correspondence: hossain.chowdhury@makautwb.ac.in

**Definition:** Antifungal drugs prevent topical or invasive fungal infections (mycoses) either by stopping growth of fungi (termed fungistatic) or by killing the fungal cells (termed fungicidal). Antibiotics also prevent bacterial infections through either bacteriostatic or bactericidal mechanisms. These microorganisms successfully develop resistance against conventional drugs that are designed to kill or stop them from multiplying. When a fungus no longer responds to antifungal drug treatments and continues to grow, this is known as antifungal drug resistance. Bacteria have an amazing capacity to become resistant to antibiotic action as well, and the effectiveness of the scarce antifungal arsenal is jeopardised by this antibiotic resistance, which poses a severe threat to public health.

**Keywords:** antifungal drugs; drug resistance; fungal infections; regulation; mycoses

## 1. Introduction to Fungal Infection

More than one billion people worldwide suffer from fungus-related illnesses known as mycoses each year, but their contribution to the world's disease burden is mostly unrecognized [1]. In 2020, an estimated 1.7 million fatalities from fungal infections were reported [2]. Healthcare practitioners have a tremendous dilemma in choosing anti-fungal agents as the prevalence of fungus infections rises alarmingly. This rise is directly linked to the rise in the number of immunocompromised people as a result of changes in medical practice, such as the use of powerful immunosuppressive medications and intense chemotherapy [3]. In the human microbiota, there are microbes like *Candida* spp. that can cause opportunistic infections in healthy people and life-threatening infections (invasive candidiasis) in people with weakened immune systems, like those with HIV, cancer patients receiving chemotherapy, and people taking immune-suppressive medications [4]. In addition to opportunistic and systemic infections, individuals with underlying disorders may develop healthcare-associated infections from fungal pathogens such *Candida*, *Aspergillus*, *Fusarium* and *Mucorales*.

Systemic fungal infections frequently result from the genera *Candida*, *Blastomyces*, *Coccidioides*, *Paracoccidioides*, *Histoplasma* and *Cryptococcus*. The fourth most frequent opportunistic infection in hospitals is *C. albicans* infection [5]. Despite the use of antifungal treatments, invasive candidiasis (IC) is deadly in about 42% of instances that have been recorded. Currently, azoles like fluconazole, itraconazole, voriconazole (VOR), posaconazole and isavuconazole (ISV), polyenes like amphotericin B (AMB), and echinocandins like caspofungin, micafungin, and anidulafungin are the most often used antifungal medications for IC [6–8].

Different components of the fungal cell are affected by these antifungal substances. Azoles stop the formation of ergosterol, the primary building block of fungal membranes [7,9]. Echinocandins act by inhibiting the formation of 1,3-β-D-glucan found in the fungal cell wall, while polyenes like AMB interact with ergosterol-producing pores in the cell membrane. Resistance to *Candida* spp. has risen as a result of the progressive rise in the likelihood of infection with *Candida* and the increased use of antifungal medications [10]. The issue of antifungal resistance and its molecular underpinnings has received interest due to pharmacological shortcomings in therapies for *Candida* spp. As a result, the current chapter offers a summary of prospective anti-fungal medicines, their mode of action, and their resistance.

### 1.1. Classification of Fungal Infections (Mycosis)

Mycosis are traditionally divided into four forms:

1. Superficial infections
2. Subcutaneous infections
3. Systemic infections

### 1.1.1. Superficial Infections

These are defined as an infection that mostly affects the stratum corneum, or the skin's outermost layer, as well as the mucous membranes, nails and hair. These infections, including those caused by *Dermatophytes*, *Trichophyton* spp., *Microsporum* spp., and *Epidermophyton* spp., are among the most prevalent diseases that people experience worldwide [11]. Transmission of the fungus occurs by direct contact with infected persons, animals, soil, or termites. Globally, 20–25% of the population is thought to have superficial mycoses, and the prevalence is increasing [12]. We can better understand future epidemiologic trends and risk factors for superficial fungal infections when we are aware of the primary causative species.

Tinea capitis is a common condition of the skin that usually affects children older than six months. Tinea versicolor, an infection of the stratum corneum, is brought on by *Malassezia* spp. including *M. furfur*, *M. globosa* and *M. sympodialis*, a yeast that lives on the skin as a commensal. Onychomycoses, or nail infections, are thought to be the cause of 50% of all nail disorders and 33% of all fungal skin infections [13].

### 1.1.2. Subcutaneous Fungal Infections

The "subcutaneous" mycoses are caused by a wide variety of diverse organisms that can spread disease when implanted or otherwise introduced into the dermis or subcutis. Mycetoma, sporotrichosis, and chromoblastomycosis are the three kinds of subcutaneous mycoses [14]. They all seem to be brought on by causing trauma to the subcutaneous tissue where the etiological fungus is located.

Verrucoid skin lesions are the hallmark of the subcutaneous mycosis known as chromoblastomycosis, which typically affects the lower extremities. The disease-specific "copper penny" cells, or muriform cells with perpendicular septa, are identified through histological analysis. Chromoblastomycosis usually only affects subcutaneous tissue and seldom affects bone, tendon, or muscle [15]. Mycetoma, on the other hand, is a subcutaneous mycosis that ravages nearby bone, skeletal muscle, and tendons. It is suppurative and granulomatous. Small, visibly coloured grains or granules are discharged from sinus tracts that form as a result of mycetoma.

Sporotrichosis is the third broad category of subcutaneous mycoses. At the site of the traumatic inoculation, the infection brought on by *Sporothrix schenckii* affects the subcutaneous tissue. The infection typically spreads through the affected extremity's cutaneous lymphatic pathways [16].

### 1.1.3. Systemic Fungal Infections

Systemic mycoses are the systemic infections predominantly caused by organisms from the genera *Candida*, *Aspergillus* and *Mucor*. In addition, disseminated infections

from *Blastomyces*, *Coccidioides*, *Paracoccidioides*, *Histoplasma* and *Cryptococcus* spp. are also found [17]. Systemic mycoses enter the body by a deep focus or an internal organ such the paranasal sinuses, digestive tract, or lungs. The infection often starts in the lungs before spreading to the skin and other organs. Usually, the infection starts in the lungs before spreading to the skin and other organs. All of these organisms, with the exception of *Cryptococcus neoformans*, are dimorphic, forming as mycelia in their natural state and converting into yeast form in tissues. As a result, there is no additional discussion of the yeast infections that cause cryptococcosis in this section. Numerous lung illnesses and a localised skin affliction have been connected to *Chrysosporium parvum*, a filamentous soil saprophyte.

## 2. Overview of Antifungal Drugs and Their Mechanism of Action

Investigating the five antifungal drug classes that have been approved for use on humans requires an understanding of the structural differences between pathogenic fungus and normal cells. The creation of antifungal drugs frequently targets the mannans, glucans, and chitins, as well as a few of the enzymes of the ergosterol biosynthesis pathways that are exclusive to fungal cells [18]. The azoles, polyenes, echinocandins, pyrimidine analogues, allylamines, thiocarbamates, and morpholines are the most widely used antifungal medications, along with a few newly developed antifungal medicines [19]. Figure 1 shows these classes of drugs and their targets in fungal cells, as well as some new potential classes currently being investigated.

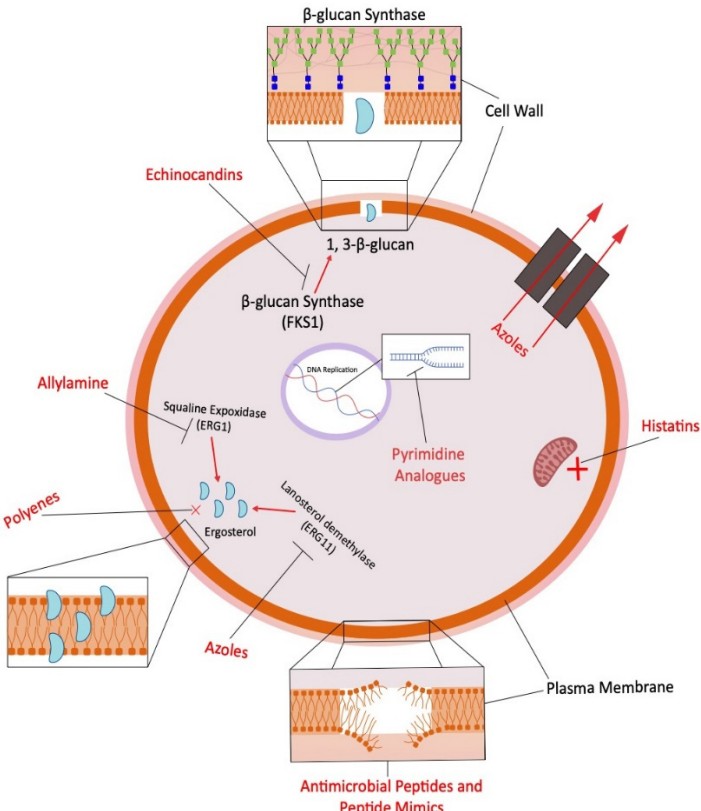

**Figure 1.** Classes of Antifungal Drugs and Their Overall Mechanism of Action. Drug class names are indicated in red.

### 2.1. Azoles

Azoles are the largest class of antifungal drugs. The first azoles to be developed were imidazole-based and included now-famous examples like miconazole (MCZ; first sold in 1974), clotrimazole (CLT; first marketed in 1975), econazole (ECO; first marketed in 1982), ketoconazole (KTC; first marketed in 1981), tioconazole (TIO; first marketed in 1983), and

sulconazole (SUL; 1989). The most recent imidazole approved is luliconazole (2013) and is used for topical treatment of dermatophytic infection. Subsequently, triazole-based drugs, including fluconazole and itraconazole were developed, exhibiting a broader spectrum of activity. Triazoles are taken for both systemic and mucosal infections, whereas imidazoles are mostly utilised for mucosal fungal infections [20]. The mechanism of azoles involve destabilizing the fungal membrane is described: The ERG11 gene encodes 14-lanosterol demethylase, which changes lanosterol into ergosterol in the fungal cell membrane. The active site of this enzyme has an iron protoporphyrin unit. The ergosterol biosynthetic route is blocked by azoles because they bind to iron. When ergosterol synthesis is suppressed, 14-methyl sterols can build up and change the stability, permeability, and function of the membrane and the enzymes attached to it [21].

### 2.2. Polyenes

Polyenes such as amphotericin B (AMB), initially isolated from *Streptomyces*, are macrolides, amphipathic organic compounds [22]. Polyenes bind ergosterol, causing pores in the plasma membrane. Ionic equilibrium is lost as membrane integrity is compromised, and cell death occurs [23]. Amphotericin B (AmpB), natamycin, and nystatin are the three major polyenes. Because of its poor absorption, natamycin and nystatin are favoured for topical infections, whereas AmpB is most effective against *Cryptococcus*, *Candida* and *Aspergillus* species in systemic invasive fungal infections [24]. However, while there is little development of resistance, drugs such as AMB are often reserved for invasive mycoses due to serious side effects.

### 2.3. Pyrimidine Analogs

The pyrimidine analogues, such as 5-fluorocytosine (5-FC) and 5-fluorouracil, are synthetic analogues of the nucleotide cytosine (5-FU). Cytidine deaminase transforms the pyrimidine analogue 5-FC into 5-FU, which is then incorporated into DNA and RNA during their synthesis and suppresses cellular activity by either preventing protein synthesis or preventing DNA replication. These medication analogues exhibit anti-*Candida* and anti-*Cryptococcus* action [23,24]. 5-FC has a high bioavailability due to its quick absorption.

### 2.4. Allylamine, Thiocarbamates and Morpholines

The antimycotics allylamines and thiocarbamates exhibit remarkably high effectiveness against dermatophytes but only moderate efficacy against yeasts. The naphthalene moiety shared by allylamines and thiocarbamates may aid in their ability to bind to the enzyme [25]. They have only mildly interacted with the mammalian enzyme responsible for synthesising cholesterol. Onychomycosis is treated with the topical antifungal drug morpholine amorolfine, which works by inhibiting the ergosterol synthesis-related enzymes 14-reductase and 7,8-isomerase [26,27]. While allylamines and thiocarbamates only affect the ERG1 gene, morpholines, which include fenpropimorph and amorolfine, block the ERG24 and ERG2 genes of ergosterol production. One of the allylamines is terbinafine, while one of the thiocarbamates is tolnaftate.

### 2.5. Echinocandins

With a focus on the synthesis of cell wall components, the lipopeptide echinocandins are a relatively new class of antifungal medications that act as non-competitive inhibitors of 1,3-β-D-glucan synthase, the enzyme necessary for β-glucan synthesis [28]. The integrity of fungal cells is impacted by errors in the production of cell wall components, leading to cell wall stress. As a result, echinocandin-treated cells exhibit separation faults, increased osmotic sensitivity, pseudohyphae formation, decreased sterol levels, and thickened cell walls [29] Echinocandins function on a certain cell wall manufacturing pathway that is peculiar to fungal cells, making them generally non-toxic to mammalian cells [30].

### 3. Pharmacological and Toxicological Study of Antifungal Drugs

In order to prevent negative effects on the healthy cells surrounding the afflicted area, effective antifungal medications must be delivered carefully. The most effective antifungal dosages are determined by the patient population and take both the patient's physical state and the type of therapy into consideration. Antifungal drugs can be difficult to prescribe for pregnant women due to a lack of research regarding its embryotoxic/teratogenic effects.

The Food and Drug Administration has designated only AMB (amphotericin B) as the safest drug for treating a systemic fungal infection. AMB has nevertheless also been linked to certain negative side effects. AMB is combined with sodium deoxycholate in the pharmaceutical formulation since it is poorly soluble in water. AMB builds up in the liver and spleen after being injected, where it clings to plasma lipoproteins and separates from deoxycholate. Due to its lengthy elimination half-life, AMB cannot be metabolised by CYP450 enzymes and is instead excreted unaltered in significant levels (33%) and (43%) in the urine and faeces, respectively [31]. AMB deoxycholate application's toxicity is limited by the fact that it is dose- and infusion-dependent. On the other hand, because renal cells are non-selectively damaged, high doses of AMB produce nephrotoxicity. To decrease these negative effects, AMB deoxycholate has been introduced in lipidic formulations that retain their fungicidal efficacy [32].

5-flucytosine, a small hydrophilic molecule, on the other hand, absorbs fast and has a bioavailability of over 90%. The amount of 5-FC that is metabolised by liver enzymes is minimal. In the bladder, it has potent antifungal activity and can only be removed via glomerular filtration. Additionally, it clears from the plasma just as quickly as creatinine does. 5-FC is dosed frequently because it has a maximum four-hour half-life. The antifungal potency of cytarabine is competitively decreased when 5-FC and cytarabine, a therapy for acute myeloid leukaemia, are given together because they share the same transport pathway as sensitive cells. Among the severe side effects are gastrointestinal problems, myelotoxicity, and liver damage [33].

Depending on their molecular weight, solubility, and capacity to bind to proteins, triazoles exhibit a variety of pharmacological properties. All triazoles can be administered intravenously or orally. Notably, isavuconazole is administered via the water-soluble prodrug isavuconazonium. The single metabolic pathway for itraconazole produces the active metabolite hydroxyitraconazole.

On the other hand, CYP2C19 and CYP3A4 genetic polymorphisms have an effect on the metabolism of voriconazole. Usually, triazoles are well tolerated. The most frequent significant adverse event associated with VOR is hepatotoxicity (in 31% cases). The other triazoles function counter to ISV, shortening the QT interval (the period of time between the beginning of ventricular depolarization and the end of ventricular repolarization). Triazoles affinity for CYP450 isoenzymes results in an unpleasant number of potential drug interactions.

### 4. Current Antifungal Therapies

The cell walls of yeast and filamentous fungi contain typical components such chitin, glucan, mannan, and glycoproteins. Since fungi are eukaryotes, as opposed to prokaryotes like bacteria, they share a lot of characteristics with their hosts' cells and metabolic processes. Drug research has been centred on the features and structures particular to fungi as a result of the biological similarity between eukaryotic hosts and fungal pathogens. Currently, systemic and superficial antifungal therapy can use five common groups of antifungal medications, including azoles, polyenes, echinocandins, allylamines and pyrimidine analogues.

#### 4.1. Polyenes

The capacity of polyenes to specifically bind sterol at the fungal cell membrane is what gives them their strong fungicidal effect. The modes of action for polyenes have been hypothesised under four different scenarios: The oxidative damage model, the sterol

sponge model, the surface adsorption model, and the pore creation model [10]. The most extensively researched mechanism involves polyenes directly intercalating with the ergosterol membrane to generate ion channels that permeabilize and kill yeast cells [34]. This process is known as the pore creation model. Reactive oxygen species (ROS) and interleukin-1 (IL-1) released by host cells are two additional indirect mechanisms of fungal cell destruction produced by polyene chemicals that have been found [35]. The first broad-spectrum fungicidal medication, amphotericin B (polyene), was created and is used to treat invasive fungal diseases, such as aspergillosis, candidiasis, blastomycosis, cryptococcosis, mucormycosis, histoplasmosis, and sporotrichosis [36]. For the initial management of a variety of uncommon fungal infections, particularly those that are severe and life-threatening, amphotericin B formulations are advised. Normally, amphotericin B inserts into the cytoplasmic membrane and binds largely to the ergosterol of the membrane. After the pore-like structure forms, the membrane's integrity is compromised and the proton gradient is abolished, which leads to ion leakage in the cell and osmotic instability [37,38]. Amphotericin B interacts more potently with ergosterol than it does with cholesterol. This differential can be used to explain why amphotericin B has a special preference for ergosterol-containing fungal membranes. The drug's association with cholesterol and its non-specific binding to lipid membranes or other macromolecules, on the other hand, are linked to apparent toxicity in patients [39]. Pore creation, however, is not a sufficient mechanism to bring about cell death. The G2/M cell cycle arrest, abnormal mitochondrial architecture, oxidative stress, and other indications of apoptosis at therapeutic levels are thought to be part of the apoptotic cascade within the cell [40]. Reports of apoptosis continue to increase [41], despite the fact that studies on the intracellular mechanism of polyenes are still not fully understood.

### 4.2. Azoles

Based on how many nitrogen atoms are in the ring, the azoles can be divided into two primary subclasses. Triazoles with three nitrogen atoms in the cyclic ring and imidazoles with two nitrogen atoms each make up the two subclasses. For cutaneous and mucocutaneous infections, the first azoles, the imidazoles (clotrimazole and miconazole), were introduced to the market in 1969 as topical substitutes for griseofulvin and one of the polyenes, nystatin [42]. The ERG11 gene-encodes 14-lanosterol demethylase, which transforms lanosterol into ergosterol in the cell membrane, is the azoles' mechanism of action. The active site of this enzyme has an iron protoporphyrin unit. The ergosterol biosynthetic route is blocked by azoles because they bind to iron [43,44]. When ergosterol synthesis is suppressed, 14-methyl sterols can build up and change the stability, permeability, and function of the membrane and the enzymes attached to it [21].

### 4.3. Echinocandins

The 1,3-β-D-glucan synthetase enzyme, which is in charge of generating 1,3-β-D-glucan, one of the main elements of the fungal cell wall, is blocked by echinocandins, lipo-peptides. This results in osmotic instability, which kills the fungal cells. Caspofungin, micafungin, and anidulafungin are all members of this class. Different lipopeptides have been demonstrated to exhibit antibacterial, antifungal, anti-adhesion, quorum sensing, or anticancer actions. These lipopeptides, a crucial part of the fungi's cell wall, have an effect on glucans demonstrating surfactant activity [45]. The fungal cell wall, which is mostly made of glucan and chitin, is thought to possess a special characteristic that protects against environmental stressors and host immunity. These elements are essential for shielding cells from osmotic pressure and environmental stress, and they are crosslinked to maintain cell integrity and form.

### 4.4. Use of Drug Combinations

Synergistic combination therapy improves therapeutic effectiveness and slows the development of drug resistance when compared to single drug-based monotherapy. Nu-

merous in vitro and in vivo investigations have demonstrated the efficacy of combinatorial methods in the treatment of fungal infections [46]. This practical approach focuses on dismantling biofilms and avoids the pipeline, which is expensive and time-consuming for the development of novel drugs. Combinatorial techniques use pharmacological combinations that have various biological targets to produce synergistic efficacy, or they increase bioavailability.

In the case of treating cryptococcal meningitis, flucytosine and amphotericin B combination therapy had a greater incidence of recurrence and less frequent relapses than amphotericin B monotherapy. The expensive price tag and unfavourable side effects of antifungal medications place restrictions on their combination. Other non-antifungal medication types, such as calcineurin inhibitors, heat shock protein 90 (Hsp90) inhibitors, calcium homeostasis modulators, and conventional therapies, may also be used with fluconazole. It is noteworthy that a few of these combinations also had a synergistic effect on bacterial strains that were resistant to common antibiotics. The primary mechanisms of this synergistic action include a rise in cell membrane permeability, a decrease in antifungal drug efflux, a disturbance of intracellular ion homeostasis, inhibition of protein and enzyme activity necessary for fungal survival, and prevention of biofilm formation. Despite their challenges, successful therapy with combination medicines should ultimately encourage scientifically rigorous, evidence-based research for evaluation [47].

*4.5. Novel Antifungal Agents*

Newer antifungals typically require extensive research and development over a long period of time and have longer schedules. This issue can be resolved by reusing non-antifungal drugs from the authorised drug library. This is done by mining and screening compounds with potential antifungal effect using computational modelling or docking techniques, followed by experimental validation [48]. Recent research utilising this technology have revealed possible antifungal action with well-known drugs such as the anti-cancer medicine tamoxifen, the anti-rheumatic drug auranofin, the calcium channel blockers nisoldipine, nifedipine, and felodipine, the anti-inflammatory drugs aspirin, ibuprofen, and atorvastatin, as well as the anti-inflammatory drug nifedipine [49]. Additional repurposed drugs, such as aripiprazole, miltefosine, benzimidazoles, quinacrine, robenidine, raltegravir, cisplatin, and pitavastatin, have been shown in animal models of fungal infection to limit the induction of hyphal growth and the creation of biofilms while simultaneously giving therapeutic advantages [49].

Strategies using natural extracts from invertebrates and plants also have yielded substances that have the potential to treat *Candida albicans* and other yeast species. The earthworm, *Dendrobaena veneta*, produces a coelomic fluid that is active against *C. albicans* [50]. When a protein-polysaccharide fraction was isolated, it was found to increase the expression of superoxide dismutase I, an oxidative stress protein, and mitochondrial proteins involved with apoptosis in yeast cells. This fraction also interrupted cell division and led to the formation of several morphological deformities of the yeast cell, including pH changes and mitochondrial DNA migration and fusion with nuclear DNA in the nucleus, ultimately resulting in the death of the cell and disruption of the cell wall. The earthworm fraction also had anti-tumour effects on A549 and CaCo cell lines while leaving normal human fibroblasts undamaged. Many other natural compounds targeting *C. albicans* mitochondria, along with excess free radical generation have been described. These include (+)-medioresinol from *Sambucus williamsii* [51,52], berberine from the herb, *Berberis vulgaris* [52], garlic allyl alcohol from *Allium sativum* [53] baicalin [54], curcumin from the plant *Curcuma longa* [55,56] and shikonin isolated from *Lithospermum erythrorhizon* [57,58].

Antimicrobial peptides (AMPs) are a naturally occurring class of host defense molecules that exhibit potent in vitro and in vivo activity against a wide variety of microbes, including fungi [59]. Identified in almost all species, they are for the most part highly cationic, and are found as either linear peptides, or as disulfide-bonded β-sheets. Their primary means of action is the rupture of bacterial, fungal, or viral membranes by electrostatic and

hydrophobic contact, which permits the leaking of intracellular contents [60]. Because AMPs target the cell membrane and repair for damaged membranes is ineffective, this physicochemical mechanism prevents the development of resistance. However, some of these peptides, notably the antifungal histatins, found in human saliva, use other mechanisms of action, including targeting mitochondrial functions [61]. Peptides with strong antifungal activity are often found in plants, and include the antifungal peptides HsAFP1, RsAFP1 and RsAFP2 [62]. These peptides have been shown to exhibit synergistic activity with conventional antifungal agents, such as caspofungin and amphotericin B, in both planktonic and biofilm forms [63]. While these peptides exhibit strong potential as new antifungal therapies, they suffer from some drawbacks, including difficulty to produce, and inactivation by proteases. However, strategies have been developed to design novel mimetics of these peptides that are insensitive to proteases, and which have been shown to exhibit activity against both oral and disseminated *Candida* infections in mouse models [64–66].

## 5. Antifungal Resistance

Antifungal resistance happens when a medicine used to treat a fungal infection is no longer effective. Some fungal species are inherently resistant to being treated with specific antifungal medications [67]. Fluconazole, for instance, has little effect on infections brought on by *Aspergillus fumigatus*. Additionally, when fungi are exposed to antifungal medications over time, resistance might form. This resistance can develop even when antifungal medications are administered correctly to treat patients who are ill (for example, when dosages are too low or treatment durations are too short).

### 5.1. Overview of Causes of Antifungal Resistance and How to Tackle the Problem

Resistance to conventional antifungal drugs is a major public health issue worldwide [68]. Antifungals have no effect on some fungi. They have a built-in resistance to multiple antibiotics. This is known as natural or intrinsic resistance. There are several reasons which affects fungal pathogens to develop resistance to standard treatments, which is known as acquired resistance. The development of antifungal resistance severely restricts the therapeutic choices available. In addition, research has shown that when a fungal pathogen develops resistance to one class of antifungal, the remaining alternatives may be less successful as well [69].

Acquired antifungal resistance has developed in therapeutically significant fungi as a result of the regular and preventive use of antifungal drugs. These have all been connected to drug target modification or overexpression, elevation of multidrug transporters, and stress response activation as adaptive mechanisms of antifungal drug resistance [70]. Although resistance to polyenes is still quite rare, resistance to azoles and echinocandins is well established, and the mechanisms of acquired resistance will be discussed in greater detail below.

Antifungal resistance can be acquired by a variety of factors, including:

- Antifungal medication misuse: When doses are skipped, therapy is stopped too soon, or the dose prescribed is too low, a fungus becomes more adept at fending off the effects of the antifungal drug [71].
- Fungicide use: Fungicidal drugs are often used in agriculture, as a method to protect crops from rotting [72]. As fungi have been exposed to more fungicides under these conditions, people working closely with fungicide-treated crops may become more susceptible to antifungal-resistant fungus diseases.
- Spontaneous resistance: This occurs when a fungus ceases to respond to a previously effective medication for no apparent cause.
- Transmitted Resistance: One can transmit a contagious drug-resistant fungal illness. Even though the recipient has had never used an antifungal drug, that individual now has an illness that will not react to that particular antifungal drug [73].

- Prolonged Treatment: Some fungal diseases require prolonged treatment, and as a result, the fungus will be exposed to the antifungal drugs for an extended period, leading to the development of drug resistance [74,75].

The best method to avoid antifungal resistance is to first identify the fungal species correctly, which is extremely important to proper selection of the antifungal drug [76]. The second-best method to avoid acquiring resistance is to take antifungal medication as directed. One may create reminders so that they do not forget to take it in proper time. Still if they miss a dosage, it is advisable to contact their healthcare practitioner to find out what to do. It is usually better to take the following dosage as soon as possible.

*5.2. Types of Resistant Fungi*

*Aspergillus* and several *Candida* spp. are fungi that have developed resistance to antifungal medications. The novel species *Candida auris* is extremely resistant to antifungal medications and is contagious in medical facilities [77]. Fungi use four basic methods to thwart medication treatments:

i.    Targeted protein overexpression (azoles);
ii.   Mutations induced in targeted proteins (azoles and echinocandins);
iii.  Enhancing the production of efflux pumps and/or increasing their insertion into cell membranes (observed in azoles); and
iv.   Limited access to the target, as in the case of ergosterol sequestration (observed in polyenes).

Approximately 7% of all systemic infections caused by *Candida* spp. show impaired azole susceptibility [78]. Three methods that fungal cells exploit singly or in conjunction with one another to thwart the fungicidal effects of these medications account for resistance to the azoles:

i.    Overexpression of ERG11
ii.   Mutations in ERG11
iii.  Expression of efflux pumps.

Lanosterol 14$\alpha$-demethylase, a suspected enzyme that the azoles are thought to target, is encoded by the ERG11 gene. Numerous studies have revealed that azole-resistant *Candida* strains have increased transcriptional levels of ERG11 mRNA, which results in increased amounts of the targeted enzyme. This robust lanosterol 14$\alpha$-demethylase expression compensates for any activity loss brought on by azole inhibition [79,80].

The emergence of lanosterol 14$\alpha$-demethylase mutations in clinical isolates of *Candida albicans* is the main component of the second efficient method that fungi employ to develop azole resistance. Although there are many mutations that confer resistance to the azoles, they seem to cluster close to the heme-binding group in lanosterol 14$\alpha$-demethylase [81,82]. Only 12 of the identified changes—A61V, Y132F, Y132H, K143E, K143R, F145L, I471T, S405F, V456I, G464S, and R467K—have been proven by in vitro methods to play a part in azole resistance. These single-site mutations reduce azole susceptibility by 4- to 64-fold [83]. K143R and V456I, two of these eight changes, for instance, are highly important in the emergence of resistance.

## 6. Resistance Mechanisms of Antifungal Drugs

The development of antifungal resistance is influenced by a variety of factors. These strategies include drug target changes, sterol biosynthesis changes, target enzyme intercellular concentration reductions, and antifungal drug target overexpression [29,76]. Understanding the various antimicrobial drugs' mechanisms of action is crucial for comprehending the processes of resistance. In many situations, elucidating resistance mechanisms has permitted or improved our understanding of particular action mechanisms.

### 6.1. Assessing Host and Environmental Factors Influencing Antifungal Resistance

Clinical resistance is defined as a patient's inability to respond to an antifungal treatment after a normal dose has been administered. Antifungal resistance is a complicated process that is influenced by a variety of host and microbial variables [84]. The immunological condition of the host is crucial since fungistatic drugs must cooperate with the immune system to control and eradicate an infection. Antifungal therapy is more likely to fail in patients with considerable immunological dysfunction since the medication must fight the infection on its own without the assistance of the immune system [85]. By allowing pathogenic organisms to adhere to medical devices, creating biofilms that are resistant to treatment, indwelling catheters, prosthetic heart valves, and other surgical implants may be a factor in refractory infections. Each drug must reach the infection site at a concentration sufficient for antibacterial activity to be effective [70]. The pharmacokinetics of many medications are understood, but there is not a clear understanding of drug penetration at all infection sites. As a result, certain microbes are inadvertently exposed to medications at insufficient concentrations. As a result of this condition, fungal cells may remain in the tissues during treatment and establish subclinical reservoirs that generate new infections in the patient. Microbial resistance, which is defined as the selection of strains that may grow despite exposure to antifungals at therapeutic dosages, is the result of all these processes. These strains are a major factor in the pharmacological failures that occur during treatment. Secondary resistant strains, which develop a resistance feature or trait in an otherwise susceptible strain after drug exposure, and primary resistant strains, which are innately less sensitive to a particular antifungal therapy, are both examples of microbial resistance.

### 6.2. Molecular Mechanisms of Antifungal Resistance

Microorganisms adopt three basic tactics to counteract the fungicidal or fungistatic effects of all antifungal classes: (i) reducing the drug's concentration inside the fungal cell; (ii) decreasing its affinity for its target; and (iii) changing metabolism to balance the drug's effects. For example, the ergosterol biosynthetic pathway underwent four main alterations after azole action, which have received the most attention in studies on the molecular mechanisms of azole resistance in yeast: (i) decrease in azole affinity for their target, (ii) increase in azole target copy number, (iii) alteration of the ergosterol biosynthetic pathway, and (iv) decrease in intracellular azole accumulation. Several mechanisms of resistance are frequently coupled in certain highly resistant clinical isolates, which were taken from long-term treated patients. Due to the progressive development of new mechanisms, resistance to antifungal therapy has increased.

### Regulation of Drug Resistance Genes

Systems that include multidrug transporters and drug targets, rely on drug resistance genes. The control of these transcriptional factors is being described since it is essential for the development of antifungal drug resistance. Complex regulatory circuits involving these genetic and transcriptional regulators have been discovered by recent genome-wide research [86]. Antifungal pressure, particularly when applied over an extended period of time, will eventually cause mutations or chromosomal rearrangements in fungal cells [87]. These occurrences may have an impact on the expression of drug resistance genes, which in turn will alter the amount of acquired antifungal resistance. Therefore, it is crucial to comprehend the regulatory system that governs drug resistance in fungi.

### 6.3. Resistance to Azoles

It has been well established for decades that pathogenic fungi exhibit azole resistance through molecular pathways. Numerous drug-resistant *Candida* clinical isolates commonly exhibit these well-studied, equally significant pathways. The various azole resistance mechanisms are discussed further below.

### 6.3.1. Over-Expression of Membrane Transporters

Pathogenic yeast contains a significant number of membrane proteins. The cell membrane, vacuolar membrane, and mitochondrial membrane all contain these membrane proteins. They perform several physiological functions, such as drug efflux, drug modification, drug detoxification, environmental sensing, nutrition transport, signal transduction, and signal transduction. For instance, the ABC transporter Atm1p, which is positioned on the mitochondrial membrane, is crucial for maintaining iron homeostasis, whereas Mlt1p, which is placed on the vacuolar membrane, transports phosphatidyl choline (PC) [88]. Multiple physiological processes can also be carried out by a single membrane transporter. Recent studies, for instance, revealed that the Mlt1p transporter is in charge of transferring PC and transporting azoles into vacuoles. Methotrexate and azoles were more susceptible to *C. albicans* after Mlt1p was deleted [89]. There are two different kinds of membrane transporters that have been linked to azole resistance in fungus:

- ABC-transporters
- MFS-transporters

### 6.3.2. Erg11 (Cyp51A) Substitutions: Ergosterol Biosynthetic Enzymes Alterations

Azole drugs target the ergosterol production pathway. When ergosterol production is suppressed, 14-$\alpha$-methyl sterols accumulate and alter the stability, permeability, and activity of membrane-bound enzymes. Ergosterol is essential for the construction of the fungal cell membrane [90]. Azoles specifically target a cytochrome P450-dependent enzyme termed lanosterol 14-$\alpha$-demethylase, encoded by Erg11 in yeasts and Cyp51A in moulds [91]. Erg11/Cyp51A catalyses the oxidative removal of the 14-methyl group from lanosterol. Azole binding to the ferric iron moiety of the heme-binding site prevents the enzyme from using its natural substrate, lanosterol, which halts the production process. Codons 54 and 220 are where Cyp51A mutations that provide acquired resistance are most frequently documented [92]. Cross-resistance within the drug class is influenced by the location and kind of the modification within the protein structure, with proximity to the heme-binding site having an impact on the binding of any azole medication.

### 6.3.3. Alterations in Ergosterol Biosynthetic Enzymes

Clinical isolates of *Candida albicans* that are azole-resistant usually overexpress the ERG11 gene. This directly adds to resistance since more medicine is needed to block a target with increased abundance, which lowers sensitivity. ERG11 constitutive overexpression in clinical isolates with azole resistance is caused by a variety of processes [93]. First, the ERG11 gene can be amplified either through total chromosome duplication or by constructing an isochromosome with two copies of chromosome 5′s left arm (i(5L)), which is where ERG11 is situated [94]. Second, disruption of *C. albicans* UPC2 causes azole hypersensitivity; active mutations in the gene encoding the transcription factor UPC2 up-regulate most of the genes involved in ergosterol synthesis [95]. It is still entirely unclear what causes this overexpression or how it contributes to azole resistance in these species. The *erg3* mutations are sufficient for azole resistance in *Candida*, despite the fact that they are rarely associated with considerable levels of resistance. The *erg3* mutations, which are associated with cross-resistance to polyenes, are thought to result from the target ergosterol being depleted [96]. *Erg3* mutations have not been linked to resistance in *Aspergillus* to this point [97].

### 6.3.4. Diffusion of Drugs

The activation of membrane-associated efflux pumps is a common resistance mechanism that promotes multidrug resistance (MDR) by recognising various substances. The ATP-binding cassette (ABC) superfamily and the major facilitator superfamily (MFS) are two separate drug efflux pathways in fungi that affect azole resistance.

*6.4. Resistance to Echinocandins*

The most recent class of antifungal medications to enter clinical use is echinocandins. By blocking (1,3)-β-D-glucan synthase, which is encoded by FKS1 (and FKS2 in *Candida glabrata*), they aim at the fungal cell wall, causing severe cell wall stress and a loss of cell wall integrity [98]. Tolerance and resistance to echinocandins can also be conferred through intricate cellular circuitry that coordinates responses to cell wall stresses. Echinocandin drug tolerance and protective mechanisms like upregulating chitin production are made possible by cell wall integrity signaling, which is regulated by the protein phosphatase calcineurin, the protein kinase C (PKC), and the molecular chaperone Hsp90 [99]. Other than these there are many other factors and mechanisms responsible for the echinocandins resistance and some of them are yet to be known.

## 7. Multidrug Resistant (MDR) Fungi

Resistance to multiple antifungal drugs has increased since 2017 principally in two species: *Candida auris* and *Candida glabrata* [100]. These two species can be found to be resistant to antifungal drugs in the echinocandin class and to fluconazole, the only two first-line monotherapeutic drugs for invasive candidiasis [101]. A fraction (3–10%) of *C. auris* isolates from patients are also resistant to amphotericin B (a polyene), leaving no effective antifungal therapy against these isolates [100]. This resistance to at least one agent in each of three drug classes puts these *C. auris* isolates in the XDR (extreme drug resistance) category [101].

MDR, as defined by an isolate that is not susceptible to at least one agent in two drug classes, can also be found in *Aspergillus* species that cause invasive, chronic, and allergic aspergillosis [100]. Many *A. fumigatis* isolates are resistant to triazoles and can exhibit pan-azole resistance. *A. niger* is resistant to itraconazole and isovuconazole, which are oral azole drugs. *A. terreus* and *A. nidulans* are resistant to amphotericin B [100]. Therefore, analysis and detection of resistant *Aspergillus* species is critical to successful therapy in patients with aspergillosis.

## 8. The Need for Development of New Antifungal Drugs

Immunosuppressed patients, such as those with HIV, cancer patients, and transplant recipients, have had their lifespans extended as a result of recent medical improvements brought about by the discovery of novel medications and the development of therapeutic approaches. However, a fast increase in human invasive mycosis infections coincided with the growth in immunocompromised people, which is now a significant global public health concern [102]. Only a few numbers of antifungal medications are now licensed for use in the treatment of mycoses, and most of them have drawbacks such high toxicity and limited efficacy [103]. The fact that some fungal species are resistant to the available treatments makes it much more difficult to treat such infections. Researchers have been looking for solutions to these problems since there is an urgent need for the development of novel antifungal agents or new therapy regimens.

## 9. Conclusions

We reviewed the recent advancements in methods and technology that have enhanced and are anticipated to make finding novel antifungal small compounds even easier. Screening huge libraries of synthetic small molecules or natural products for their capacity to prevent the development of a chosen fungus has historically been the most popular method of discovering antifungal small molecules. The significance of the chemical properties and place of origin of the molecules in the library have come to be more appreciated in recent years. There has been an expansion in the number of libraries of synthetic small compounds that are readily available commercially as high-throughput screening has become a tool for both drug discovery and biological inquiry. Regarding mammalian targets and physiology, the vast majority of the molecules in these libraries have been created or gathered using standards that maximize their "drug-like" properties. Unfortunately, efficient anti-infective

compounds have physicochemical features substantially different from molecules intended for other therapeutic applications; this is owed in part to the necessity that the molecule cross microbial cell walls. Therefore, libraries focused on a different set of "drug-like" qualities may be useful in fresh screening attempts for antibacterial or antifungals.

**Author Contributions:** Conceptualization, C.M.H., L.K.R. and G.D.; resources, S.C., M.G. and G.D.; writing—original draft preparation, S.C., M.G., N.L. and K.A.A.; writing—review and editing, C.M.H., L.K.R. and G.D. All authors have read and agreed to the published version of the manuscript.

**Funding:** This research received no external funding.

**Institutional Review Board Statement:** Not applicable.

**Informed Consent Statement:** Not applicable.

**Acknowledgments:** We would like to thank Allison Ryan for the Figure 1 illustration.

**Conflicts of Interest:** The authors declare no conflict of interest.

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
