# Peer review of "Antifungals and Drug Resistance"

_encyclopedia, doi:10.3390/encyclopedia2040118_

Round 1

Reviewer 1 Report

Pleases see attached file. comments put on pdf.

Author Response

Reviewer 1:

  1. We have corrected the sentence to explain that the species are the causative agents of the infection.
  2. We have included the other species of Malassezia.
  3. We have emphasized the primary contribution of Candida, Aspergillusand Mucor species in systemic fungal infection.
  4. Information on fluconazole, itraconazole and luliconazole have been added to the text.
  5. This section has been expanded with more detailed information on Amphotericin B.
  6. A reference for this statement has been added.

Reviewer 2 Report

The manuscript is a synthetic and valuable elaboration of an issue - Antifungals and Drug Resistance. The antifungal drug review is done well. The manuscript reviews advances in methods and technology for the development of new antifungal compounds. I propose to supplement the references with publications describing the action of natural new substances of animal origin - from earthworms and  from plant seeds. I present the proposed items below:

FioÅ‚ka M.J., Czaplewska P. Wójcik‑Mieszawska S. Lewandowska A.,  SofiÅ„ska‑Chmiel W. Buchwald T. (2021). Metabolic, structural, and proteomic changes in Candida albicans cells induced by the protein-carbohydrate fraction of Dendrobaena veneta coelomic fluid, Scientific Reports 11(1) doi:10.1038/s41598-021-96093-1.

The publication can be cited in  4.5. Novel Antifungal Agents.

The manuscript would be more interesting if it was enriched with some diagrams or drawings related to the anti-fungal effect.  

After the introduction of the suggested additions, the manuscript deserves to be published in the Encyclopedia journal.

Author Response

Reviewer 2:

  1. We have added a paragraph to include natural antifungal drugs as requested in lines 385-400 and have included the suggested Fiolka et al. 2021 reference as suggested, as well as other references.
  2. We have included a drawing as Figure 1, depicting mechanisms of resistance in section 2.

Reviewer 3 Report

Reviewer’s comments

In the review article titled ‘Antifungals and drug resistance,’ the authors Hossain et al. have collected information on the antifungals, their mechanism of action, and factors that lead to drug resistance.

The article is well written but needs to be organized better to make it crisp and avoid repetitions. Some of the below suggestions can improve the value of the manuscript.

  1. Line 17-18: Some of these…. are fungistatic. This sentence is almost a repetition of the previous sentence and hence may be removed. 
  2. Definition/Abstract: The authors may first describe the antifungal resistance and then the resistance by bacteria. 
  3. Line 180-183: need reference
  4. The authors may consider combining subtitles 2 (Overview of antifungal drugs and their mechanism of action) and 4 (Current antifungal therapies)
  5. Line 321: All the subtitles, such as 6, 7, 8, can be a part of subtitle 5 (Antifungal resistance). Avoid repetition.
  • Another cause of development of resistance: Some of the fungal diseases need prolonged treatment, and hence fungus will be exposed to drugs for an extended period, leading to the development of drug resistance (PMID: 28886681, PMID: 35948057), 
  • Multidrug resistance strain of the fungus may be included (PMID: 28911043PMID: 35190454)

Author Response

Reviewer 3:

  1. Line 17-18 has been removed, as suggested.
  2. In the Definition, we first described the antifungal resistance and then the resistance by bacteria, as suggested.
  3. We added a reference to lines 180-183.  This is reference [32]: Stone et al. 2016.
  4. We have combined sections. There still may be repetition between parts 2 and 4, but part 4 is current therapies and describes mechanism of action in more depth, whereas part 2 is a more general description.  Efforts to combine other sections have been made.
  5. This is the main title, Antifungal Resistance, in line 321 of the 1st submission. We have combined and rearranged paragraphs from sections 6, 7 and 8:
    - Former section 6 has been expanded upon (more written) and has become section 5.1, “Overview of causes of antifungal resistance and how to tackle the problem”. Former secton 7.1.1. “Acquired resistance” has been moved to subsection 5.1, second paragraph in the new revised document. Former subsection 5.1, “Types of resistant fungi” has become subsection 5.2 in the revised document.
    - Detailed mechanisms of resistance are moved to a new section 6, “Resistance mechanisms of antifungal drugs” and are subsections 6.1, 6.2, 6.3, 6.4. We did not put this is section 5 because we wanted section 5 to be an overview and section 6 to describe more scientific detail.
    o Former subsection 7.1 became subsection 6.1 and was retitled in the new revision, “Assessing host and environmental factors influencing antifungal resistance”.
    o Former subsection 7.2 became subsection 6.2, “Molecular mechanisms of antifungal resistance” in the new document.
    o Former section 8 became sub-subsection 6.2.1, “Regulation of drug resistance genes”.
    o Former sub-subsection 7.1.2 became subsection 6.3, “Resistance to azoles”.
    o Former bold headings under subsections 7.1.2 became sub-subsections 6.3.1, “Overexpression of membrane transporters”, 6.3.2, “Erg11 (Cyp51A) substitutions: Ergosterol biosynthetic enzymes alterations”, 6.3.3, “Alterations in ergosterol biosynthetic enzymes”, 6.3.4, “Diffusion of drugs”.                                                                                                            o Former sub-subsection 7.1.3 became subsection 6.4, “Resistance to
    echinocandins”.
    - Section 7 is new and is called, “Multidrug Resistant (MDR) Fungi”. We felt that this is so important that it deserves its own heading. This section was suggested by Reviewer 3 also in point #6.
    - The new section 8 is retitled to “The need for development of new antifungal drugs”. This was formerly section 9 “Development of antifungal drugs and recent approaches” in the original document.                             
  6. The issue of prolonged treatment has been added to new section 5.1 (formerly section 6) and the two suggested references were added. We also added a new section called Multidrug resistant (MDR) fungi, with the new references suggested by reviewer 3.

Round 2

Reviewer 1 Report

no comment.

Reviewer 2 Report

The amendments were implemented correctly.

Reviewer 3 Report

The authors have incorporated suggestions from reviewers.